# Cottonseed Protein, Oil, and Minerals in Cotton (*Gossypium hirsutum* L.) Lines Differing in Curly Leaf Morphology

**DOI:** 10.3390/plants10030525

**Published:** 2021-03-11

**Authors:** Nacer Bellaloui, Rickie B. Turley, Salliana R. Stetina

**Affiliations:** Crop Genetics Research Unit, Agricultural Research Service, USDA, 141 Experiment Station Road, Stoneville, MS 38776, USA; rick.turley@usda.gov (R.B.T.); sally.stetina@usda.gov (S.R.S.)

**Keywords:** cottonseed, near-isolines, nutrition, seed protein, seed oil, macronutrients

## Abstract

Cottonseed is an important source of protein, oil, and minerals for human health and livestock feed. Therefore, understanding the physiological and genetic traits influencing the nutrient content is critical. To our knowledge, there is no information available on the effects of leaf shape—curly leaf (CRL)—on cottonseed protein, oil, and minerals. Therefore, the objective of the current research was to investigate the effect of the curly leaf trait on cottonseed protein, oil, and minerals in cotton lines differing in leaf shape. Our hypothesis was that since leaf shape is known to be associated with nutrient uptake, assimilation, and photosynthesis process, leaf shape can influence seed protein, oil, and minerals. A two-year field experiment using two curly leaf lines (Uzbek CRL and DP 5690 CRL) and one normal leaf (DP 5690 wild type) line was conducted in 2014 and 2015 in Stoneville, MS, USA. The experiment was a randomized complete block design with three replicates. The results showed that both Uzbek CRL and DP 5690 wild type lines had higher seed oil, and nutrients N, P, K, and Mg than DP 5690 CRL. Calcium was higher in DP 5690 CRL for two years and protein was only higher than the parents in 2015. Consistent significant positive and negative correlations between some nutrients were observed, which may be due to environmental conditions, especially heat. This indicates that curly leaf trait may partially regulate the accumulation of these nutrients in seeds. The results demonstrated that leaf shape trait—curly leaf—can affect cottonseed nutritional qualities. This research is important to breeders for cotton selection for high seed oil or protein, and to other researchers to further understand the genetic impact of leaf shapes on seed nutritional quality. It is also important for scientists to use leaf shape as a tool for physiological, biochemical, and morphological research related to leaf development.

## 1. Introduction

Cottonseed is among the major sources of nutrients for human nutrition and livestock feed. Cottonseed contains protein, oil, carbon (C), nitrogen (N), sulfur (S), and minerals, including phosphorus (P), potassium (K), calcium (Ca), and magnesium (Mg) [1,2,3,4]. Therefore, understanding morphological, physiological, and genetic traits contributing to high level of these nutrients to maintain high quality cottonseed is essential. In the current research, a leaf shape trait was used to investigate its effects on cottonseed protein, oil, and minerals in near-isogenic lines differing in leaf shape (curly leaf). Previous research reported that the curly leaf gene(s) is involved in the cell elongation and division during leaf morphogenesis. For example, the authors of [5,6,7] studied the development of leaf cells of wild type *Arabidopsis thaliana* and of the curly leaf mutant. The curly leaf mutant *Arabidopsis thaliana* was characterized and it was found that it had normal roots, hypocotyls, and cotyledons, reduced dimensions of leaves and the stems, and a decrease in cell elongation and cell number [5]. In addition, it was reported that the period of leaf development was similar to both the mutant and wild type, but the rate of cell elongation and cell division were lower in the curly leaf mutant. Morphological studies showed that multicellular organisms’ morphologies are regulated by mechanisms that control shapes, sizes, and numbers of cells [5], and the rates of cell division and elongation are known to contribute to the final shape of the leaf [8]. However, the pattern of leaf blade growth and morphology are quite complex [9,10]. Kim et al. [5] found that the two-dimensional growth of the leaf blade was genetically controlled by regulation of the polar elongation of cells, supporting previous research by others [11]. Kim et al. [5] reported that the curly leaf gene in *Arabidopsis thaliana* is required for stable repression of the floral homeotic gene, *AGAMOUS*, in leaves and stems, and the curly leaf phenotype could be due to the result of misexpression of AG. Recent studies also suggest that the wild type curly leaf gene is required to repress transcription of the AG gene in leaves, inflorescence stems and flowers [7,12]. 

It was reported that leaf shape alleles in cotton serve as a model for leaf development research and production benefits [13]. They found that leaf shape had consistent effects on boll rot resistance, earliness, flowering rate, chemical spray penetration, lint, and yield. However, research on the effects of leaf shape on various insect resistances, photosynthetic rate, water use efficiency, and cotton fiber quality were not consistent. This was due to canopy closure and light harvesting, affecting the photosynthesis process [13], and consequently nutrient uptake, assimilation and organic compound synthesis such as protein oil, and various nutrients in leaves, seeds, and fiber. 

Understanding the molecular processes controlling leaf shape phenotypic changes may help advance our efforts in developing cotton cultivars with ideal leaf shapes, enhancing sustainability, profitability, and production [13]. Morphological, physiological, molecular, and genetic research was conducted to understand the effect of leaf shape on cell division and elongation, leaf development, and cottonseed and fiber production and quality. For example, it was reported that major leaf shapes of Upland cotton are a multiple allelic of a single incomplete dominant genetic locus *L-D1* on chromosome 15-D1 (Chr15). Major leaf shape genes were mapped in cotton and gene expression changes, resulting in leaf shape phenotypic diversity. Additionally, it was found that, using cultivars of normal leaves or broad leaves (NL), subokra/Sea-Island (subOL), Okra (OL), and Superokra (superOL), leaf shapes affected cotton disease and insect resistance, as well as yield production and fiber quality. It was found that okra leaf cultivars had lower yields rot by 7–11% due to boll compared to normal leaves when boll rot level was low to moderate severity [14,15], and 43–45% when boll rot was severe [16]. In addition, superokra leaf cultivars showed further reduction under disease infection, resulting a reduction in boll rot by 55% than normal leaf cultivars under severe conditions [17]. It was explained that the reduction in boll rot was due to microclimatic differences that led to canopy closure/openness, allowing for greater air circulation and light penetration. These conditions provide less favorable environment for microbial growth and activities [17]. Additionally, it was found that Okra leaf cultivars were earlier genotypes (earlier to flowering and maturity) than normal leaves cultivars [14,18]. The four-leaf shapes are well-established as alleles at a single locus called *L* or *L-D1*. The earlier maturity (increase in flowering rate and decrease in time to maturity) was due to the fact that okraleaf cultivars and superokra leaf cultivars had less photosynthate for development of new leaves and maintenance of existing leaves, and contribute more assimilate to reproductive development earlier [19]. In addition, these cultivars initiate more bolls than they can support, leading to higher rates of abortion [19]. Using near-isogenic lines showed that okra leaf cultivars had less damage from the pink bollworm (PBW) *Pectinophora gossypiella* [20,21]. Again, the reduction in this infection was partially due to earliness of okra leaf cultivars [20]. In another study using leaf shape NILs, it was shown that only three genetic backgrounds out of seven had a reduction in PBW damage by okra leaf genotypes [22]. Another study showed that Stoneville 7A-Okra was 50% damage from PBW than its NIL [20,22] due to decreased boll penetration by PBW larvae, and Stoneville 7A-Okra leaf genotypes had increased thickness of the carpel (boll) walls in some genetic backgrounds [22].

Liu et al. [23] reported that curly leaf gene, a histone methyltransferase of Polycomb Repressive Complex 2 (PRC2) for trimethylation of histone H3 Lys 27 (H3K27me3), is thought to be a repressive regulator controlling mainly postgermination growth in *Arabidopsis thaliana*, and about 14 to 29% of genetic regions are represented by H3K27me3 in the *Arabidopsis* genome. However, transcriptional repression activities of PRC2 is still not understood. Using transcriptome profile analysis, they showed that about 11.6% genes in the *Arabidopsis* genome were repressed by curly leaf in some plant organs. They also showed that about 54% of these genes were repressed in siliques, suggesting the involvement of H3K27me3 in embryonic development. For example, plants of curly leaf 28 produced bigger sized and higher weight seeds with higher oil contents, larger oil bodies, and altered long-chain fatty acid compositions compared with wild type. This shows that curly leaf silences specific gene expression modules and genes involved in these modules, contributing to the regulation of specific physiological function during embryo development. Katz et al. [24] showed that the *Arabidopsis* FERTILIZATION-INDEPENDENT ENDOSPERM (FIE) polycomb group protein regulated the development of endosperm and embryo and repressed the flowering during embryo and seedling development. They found that low level of FIE resulted in significant alteration of plant morphology, including loss of apical dominance, curled leaves, early flowering and homeotic conversion of leaves, flower organs, and ovules into carpel-like structures. They reported that these morphological changes were similar to those exhibited by the overexpressing *AGAMOUS*. They concluded that FIE is essential for the control of shoot and leaf development [24]. Therefore, these morphology changes could be due to the differential expression of alleles of the same genes in a parent-of-origin-specific manner [25]. They also reported that high-throughput sequencing analyses showed that more than 200 loci were imprinted in *Arabidopsis*, and most imprinted loci were characterized as maternally expressed genes (MEGs); PHERES1 (PHE1) and ADMETOS (ADM) are paternally expressed imprinted genes (PEGs). Genomic imprinting is the parent-of-origin-dependent differential allelic expression of a single gene. Jeong et al. [25] reported that a gene encoding an E3 ligase (UPWARD CURLY LEAF1 (UCL1)) that degrades the CURLY LEAF (CRL) polycomb protein is a PEG. They found that after fertilization, paternally inherited UCL1 was expressed in the endosperm, but not in the embryo, and polycomb repressive complex 2 (PRC2) silences the maternal UCL1 allele in the central cell prior to fertilization and in the endosperm after fertilization. The UCL1 imprinting pattern was not affected in paternal PRC2 mutants. They also found that the maternal UCL1 allele is reactivated in the endosperm of *Arabidopsis* lines with mutations in cytosine DNA METHYLTRANSFERASE 1 (MET1) or the DNA glycosylase DEMETER (DME), which antagonistically regulate CpG methylation of DNA. On the other hand, maternal UCL1 silencing was not altered in mutants with defects in non-CpG methylation. They concluded that silencing of the maternal UCL1 allele was regulated by both MET1 and DME.

Mineral nutrition in plants is essential for plant growth, development, production, and seed quality. Deficiencies in minerals at any plant stages result in yield loss and poor seed quality. Physiological and biochemical roles of macronutrients such as S, Ca, K, Mg, and P, or micronutrients such as Fe, B, and Zn have been previously reported for plants [26,27]. For example, the roles of K^+^ [28] and Ca^2+^, Cl^−^, and Na^+^ [29,30] in osmotic pressure and regulation of stomatal opening [28] and cell membrane integrity and function [31] have previously been shown. The role of K in protein synthesis, glycolytic enzymes, photosynthesis, cell expansion and turgor, carbohydrate movement, stomatal regulation, osmoregulation, energy status, charge balance, homeostasis [27,32], and transpiration [33] were also explained. Phosphorus involvement in several physiological and biochemical processes such as nucleic acids, phospholipids, phosphoproteins, energy storage and transfer, photosynthesis, and enzyme regulation were reported by others [27,34]. P has roles in stomatal conductance [35], photosynthesis [36], cell membrane stability, water relations [37], solute movement, stomatal function, signaling systems [38], osmoregulation [28], and Ca^2+^ ATPases to restore and maintain homeostasis by pumping Ca^++^ out of the cytosol to terminate a signaling event [39]. 

The objective of the current study is to investigate the effects of leaf shape (curly leaf trait) on cottonseed protein, oil, C, N, S, and minerals (macronutrients) using near-isogenic lines differing in leaf shape. Our hypothesis was that leaf shape will influence cottonseed nutrients as leaf shape is known to be associated with nutrient uptake, assimilation, and the photosynthesis process.

## 2. Results and Discussion

An ANOVA showed that line and year were significant for protein, oil, N, S, and minerals (P, K, Mg, and Ca) (Table 1). Year × line interactions were significant for protein, oil, Mg, K, and N, but not for Ca, P, S, and C, indicating that the rankings differ in each year for protein, oil, Mg, K, and N, but not Ca, P, S, and C. In 2014, mean values of nutrients showed that Uzbek CRL and DP 5690 wild type had higher seed oil, N, P, K, and Mg than DP 5690 CRL, but protein content in DP 5690 CRL was higher in 2015 only or high as in DP 5690 wild type in 2014, although Uzbek CRL was higher than both lines (Table 2). In 2015, protein in the DP 5690 CRL was higher than both Uzbek CRL and DP 5690 wild type, but oil was higher in Uzbek CRL and DP 5690 wild type than the DP 5690 CRL line. The contents of N, C, and other minerals were higher in Uzbek CRL and DP 5690 wild type than DP 5690 CRL, similar responses as in 2014 (Table 2 and Table 3).

In 2014, the correlation showed that there were negative significant correlations between Ca and oil; K and Ca; P and Ca; S and Ca; C and Ca (Table 4). A positive significant correlation was shown between K and oil; Mg and oil; oil and P; oil and S; oil and C; oil and N. Potassium showed positive correlations with Mg, P, S, C, and N. Additionally, positive correlations were shown between Mg and S and between Mg and C; between P and S, between P and C, and between P and N; between S and C, between S and N; between C and N.

In 2015, a negative correlation was shown between protein and oil; between protein and K, P, S, C, and N (Table 5). A negative correlation was also found between oil and Ca; between Ca and K, P, S, C, N. A positive correlation was recorded between K and P, S, C, and N; between P and S, C, and N; between S and C, and N; between C and N.

Since there is no literature available on the effects of curly leaf trait on seed protein, oil, and minerals in cotton, we reported related topics to our research in this discussion. The significant effects of line and year indicated that both of these factors are important for nutrient contents. The significant interaction effects (line × year) for some nutrients such as protein, oil, Mg, K, and N may reflect that growth conditions were different in each year due to temperature or drought. Since the experiment was irrigated, the effect of drought could be minimally compared with temperature. The significant effects of year × line indicated that the nutrients had different ranking, may be due mainly to temperature changes in each year. For example, in 2014, the maximum temperatures were 31.39, 31.15, and 32.49, and 31.28 °C, respectively, in June, July, August, and September; in 2015 the maximum temperatures were 32.63, 34.10, 33.35, and 33.0 °C, respectively, in June, July, August, and September (Figure 1) [40].

Therefore, 2015 was warmer than the 2014 growing season. It was shown that growing crops under different environmental conditions, especially under high temperatures, particularly during the crucial seed-filling period, can affect growth and alter seed nutrient levels, including protein, oil, and fatty acids profiles [41,42,43,44,45,46,47,48]. The higher oil in the Uzbek line and DP 5690 wild type than DP 5690 CRL could be partially due to the fact that leaf shape affected oil content and yield. Unpublished data showed that Uzbek CRL and DP 5690 wild type had larger seeds than DP 5690 CRL, and the highest 100-seed weight was recorded in the Uzbek CRL line (12.04 g) compared with the DP 5690 wild type (10.43 g) and DP 5690 CRL (9.48 g). The Uzbek CRL line had the largest boll, followed by the DP 5690 wild type, and then DP 5690 CRL, which had the smallest size. Figure 2 shows the size of the three lines in the field, and the bolls for both Uzbek curly leaf and DP 5690 curly leaf. 

Previous research showed that leaf shape alleles in cotton had production benefits and had consistent effects on boll rot resistance, earliness, flowering rate, chemical spray penetration, lint, and yield [13]. They also indicated that leaf shape effects on insect resistances, photosynthetic rate, water use efficiency, and fiber quality were not consistent. They explained that leaf shape benefit effects were due to canopy closure and light harvesting, influencing the photosynthesis process, nutrient uptake, assimilation, and organic compound synthesis such as protein, oil, and various nutrients in leaves, seeds, and fiber [13]. In addition, the canopy of a curly leaf line is different from that of a normal leaf line, resulting in more light penetration in curly leaves. However, in normal leaf lines, it might take more time for the lower part of the canopy to heat up (due to shading) and it also might be slower to cool off. These leaf shape parameters could also be a source of seed composition and mineral nutrient differences between curly and normal leaves. It was also explained that the reduction in boll rot was due to a microclimate differences and canopy closure/openness, allowing for greater air circulation and light penetration, providing a less favorable environment for microbial growth and activities [17]. Although Wilson and George [21] reported an 8% reduction loss due to okra leaf and Landivar et al. [19] reported a 5% yield reduction, okraleaf genotypes can perform well under optimal growth environments and okra leaf cotton provided greater reproductive structures compared with normal leaf [40]. In addition, subokra lines were shown to have a higher yield than normal leaf by 3.0% [49], especially in better growth environments that allow greater plant stature. Others reported that the earlier maturity of Okraleaf cultivars and superokra leaf cultivars had less photosynthate for development of new leaves and maintenance, but more assimilate for the earlier reproductive development [19]. Our research showed that both Uzbek CRL and DP 5690 wild type had higher seed oil, N, P, K, and Mg than DP 5690 CRL. The decrease in these nutrients in DP 5690 CRL compared with the parents could be due to the fact that introducing leaf curly shape trait/gene resulted in alteration uptake, assimilation, and metabolism of protein and oil and other nutrients. This is clearly shown by the higher level of cottonseed oil and nutrient accumulation in the parent lines compared to DP 5690 CRL. Our preliminary results on cotton lint also showed that cotton lint of both Uzbek CRL and DP 5690 wild type had more N, P, K, and Mg than DP 5690 CRL (data not shown). However, Ca content was higher in DP 5690 CRL than Uzbek CRL or DP 5690 wild type line. 

Previous researchers explained that leaf shape genes are involved at the morphological, physiological, molecular, and genetic levels. For example, Liu et al. [23] reported that curly leaf gene, a histone methyltransferase of polycomb repressive complex 2 (PRC2) for trimethylation of histone H3 Lys 27 (H3K27me3), is thought to be a repressive regulator mainly controlling postgermination growth in *Arabidopsis thaliana*, and about 14 to 29% of genetic regions are represented by H3K27me3 in the *Arabidopsis* genome. Although the transcriptional repression activities of PRC2 are still not understood, about 54% of these genes were repressed in siliques, suggesting the involvement of H3K27me3 in embryonic development. They showed in their study that plants of curly leaf 28 produced bigger sized and higher weight seeds with higher oil contents, larger oil bodies, and altered long-chain fatty acid compositions compared with wild type. They concluded that curly leaf silences specific gene expression modules and genes contribute to regulate specific physiological function during embryo development. Jeong et al. [25] reported that the morphology changes could be due to the differential expression of alleles of the same genes in a parent-of-origin-specific manner [25]. Our findings support those of Liu et al. [23] in that seeds of Uzbek CRL and DP 5690 CRL leaves had higher total oil, although we have no evidence as to whether or not the oil profile was altered. Further research may be needed to find out about oil profile (palmitic, stearic, oleic, linoleic, and linolenic fatty acids). Our previous research on NILs differing in seed fuzz [3,4] or leaf color [45] showed that seed oil was higher in fuzzless genotype, but Ca and C proteins were higher in fuzzy genotypes. Additionally, N, S, B, Fe, and Zn were higher in most of the fuzzy genotypes [3,4]. Using NILs differing in leaf color, Bellaloui et al. [45] found that, generally, green leaf lines had higher contents of oil than yellow leaf lines. Additionally, seed C, N, P, B, Cu, and Fe contents were higher in green lines than in yellow lines, and there were significant correlations between protein and nutrients, and between oil and nutrients in only one year as the temperature was warmer in one year than the other [45]. Therefore, morphological traits such as leaf shape can influence the regulation of carbon and nitrogen metabolism, and uptake and assimilation of nutrients, especially those involved in the photosynthesis process and oil production such as N, C, S, P, K, and Mg, supporting the findings of others [3,4,13,23,25,45]. 

Based on the above observation on the correlation between nutrients, it was clear to see that some correlation trends were consistent over years and some were not. For example, the negative correlation between oil and Ca; between Ca and K; between Ca and P; between Ca and P; between Ca and S; between Ca and C were shown in both years (2014 and 2015). However, the negative correlation between oil and K; between oil and Mg; between oil and P; between oil and S; between oil and C; between oil and N was only observed in 2015. On the other hand, a positive correlation was only shown in 2014 between oil and K; between oil and Mg; between oil and P; between oil and P; between oil and C; between oil and N. Sometimes, there was no correlation between nutrients—for example, in 2014, protein did not show any correlation between nutrients, but in 2015, protein showed a negative correlation with all nutrients, except Ca and Mg. The inconsistency or changes of the correlation trends (positive, negative, or no change) of some nutrients mainly depend on growth conditions of the crop in each year, especially heat and drought, although drought may play a minor role if the field is irrigated. Conducting a correlation within each separate correlation analyses on curly leaf lines and normal leaf lines for nutrients did not result in additional information (data not shown). For example, in 2015, there were no correlations between nutrients in in DP 5690 CRL and Russian CRL, and only positive correlations were seen between protein and N, and between Mg and S in Russian CRL. In 2014, there were only correlation between Mg and S; between P and N in DP 5690 CRL; between protein and Ca; between Mg and C in Russian CRL; between Mg and N; between P and C in wild type DP 5690. The few correlations between nutrients, when separate correlation analyses were conducted, was due to a few data points involved in the correlation (number of observation for Pearson correlation on separate correlation is 3 versus 9 across all lines). Therefore, correlation across lines resulted in additional information (Table 4 and Table 5).

Positive and negative correlations between nutrients were previously reported and the correlations depend on growth conditions, genotype, and nutrient supply [26,27]. Additionally, it was also reported that nutrient uptake, translocation, redistribution, and accumulation processes control the accumulation of nutrients in seeds [50,51], and most of these processes and their genetic bases are still not understood [52]. The inconsistency of the correlation between nutrients across years was observed by other researchers in cotton and others crops in that the correlation can change from positive to negative to no-change, depending on the year, and was attributed mainly to gene x environment interactions [3,4,41,42,43,44,45,46,47]. Further research is needed to understand the nature of the nutrient relationship as this relationship (negative, positive, or no-change) between nutrients is important because it determines nutrient uptake, assimilation, and metabolism, which impacts production and seed nutritional quality.

## 3. Materials and Methods

Cultivar DP 5690 (Bayer Corporation, Whippany, NJ, USA; PVPC 009100118) was developed into DP 5690 F_6_ by single seed descent for six generations. After three years of single seed descent, DP 5690 F_6_ (wild type) was obtained with a theoretical purity of 98.44%.

The DP 5690 wild type was crossed with the Uzbek CRL, and the F_1_ progeny were self-pollinated in the greenhouse once. DP 5690 wild type was backcrossed to F_2_ plants with curly leaves. This process was repeated through six generations from BC_1_F_1_ through BC_5_F_2_, and resulted in normal and curly leaf phenotypes in a near-isogenic DP 5690 background. A cotton line (*Gossypium hirsutum* L.) expressing the curly leaf phenotype was obtained from a collection of cotton lines originating in Uzbekistan and was sent to the USDA Agricultural Research Service in Stoneville, Mississippi (W. Meredith, personal communication). 

A two-year field experiment was conducted in Stoneville, MS, in 2014 and 2015 to investigate the effect of leaf shape trait (in our case curly leaf, CRL) on cottonseed protein, oil, and macronutrients, including carbon (C), nitrogen (N), phosphorus (P), potassium (K), calcium (Ca), magnesium (Mg) in a DP 5690 cotton (*Gossypium hirsutum* L.) background. Two near-isogenic lines (DP 5690 curly leaf and DP 5690 wild type) and the Uzbek curly leaf parent were used. Single-row field plots with 1.02 m apart and 8.53 m in length were planted with in a Bosket very fine sandy loam soil (fine loamy, mixed, active, thermic Mollic Hapludalfs) (Soil Survey Staff, 2014) on 5 May 2014 and 30 April 2015. Pentachloronitrobenzene (Terraclor Super × 18.8 G, Chemtura USA Corporation, Middlebury, CT, USA) was applied in furrow at 11.2 kg/ha to manage seedling diseases. Field management and standard agronomic practices for cotton production were used as recommended by the Mississippi Delta region [53]. Furrow irrigation was used. Plots were treated with defoliant (thidiazuron and diuron; Ginstar EC, Bayer CropScience, Research Triangle Park, NC, USA) and boll opener (ethephon; Boll Buster, Loveland Products, Inc., Greeley, CO, USA) to maximize boll opening, cotton fiber quality and yields. The boll samples were collected by hand on 6 October 2014 and 2 October 2015, and seed cotton was processed on a standard saw gin. Seeds were acid-delinted prior to protein, oil, and nutrient analyses.

### 3.1. Experimental Design and Statistical Analysis

The experiment was a randomized complete block design with three replicates. Analysis of variance was conducted by PROC MIXED in SAS (SAS, SAS Institute, 2002–2010, Cary, NC, USA) [54]. Year and genotype were considered as fixed effects. Rep (Year) was considered as a random factor. The residuals refer to Restricted Maximum Residual Likelihood (REML) values [3,4], which reflect the total variance of the random parameters in the model. Means were separated using Fisher’s protected least significant difference test at significant level of 5% using SAS (SAS Institute, 2002–2010, Cary, NC, USA) [54]. Correlations were conducted using PROC CORR in SAS. Since year by genotype interactions were significant for some seed composition constituents, and results were presented by each year.

### 3.2. Soil Nutrient Analysis

Random samples were collected from across the field in 2014 and 2015. Soil analysis for nutrient levels in soil was conducted by inductively coupled plasma spectrometry (Thermo Jarrell-Ash Model 61E ICP and Thermo Jarrell-Ash Autosampler 300 (C Jarrell-Ash Corporation, Waltham, MA, USA)) as previously detailed [55]. Briefly, a sample of 5 g soil:20 mL Mehlich-1 solution was used for analysis. Analysis of N, S, and C were based on the Pregl-Dumas method [56,57] using a C/N/S elemental analyzer with thermal conductivity cells (LECOCNS-2000 elemental analyzer, LECO Corporation, St. Joseph, MI, USA). Oxygen atmosphere at 1350 °C was used to combust soil samples and to convert elemental N, S, and C into N_2_, SO_2_, and CO_2_ gases. The content of N, S, and C in soil was analyzed by the elemental analyzer as previously detailed (Bellaloui et al., 2015b). Soil analysis showed no nutrient deficiencies in soil. The following are averages of nutrient content in soil across the field: C = 1.02%, N = 0.11%; (g∙kg^−1^) P = 0.289, K = 2.13, S = 0.084, Ca = 4.1, Mg = 3.0, and Fe = 21.03; (mg∙kg^−1^) B = 1.8; Cu = 15.2; Zn = 62.5. Organic matter in soil was 2.87%. The crop did not show any nutrient deficiency symptoms under these conditions.

### 3.3. Analysis of Seed Minerals, N, S, and C

Nutrients N, S, and C were conducted in the ground, dried samples. Seed samples were ground with a Laboratory Mill 3600 (Perten, Springfield, IL, USA) and analyzed by digesting a 0.6 g in HNO_3_ in a microwave digestion system and nutrients were quantified using inductively coupled plasma spectrometry (Thermo Jarrell-Ash Model 61E ICP and Thermo Jarrell-Ash Autosampler 300) [55]. Seed N, C, and S were determined by the C/N/S elemental analyzer as detailed previously [47,55].

### 3.4. Determination of Seed P

Phosphorus concentrations in mature seeds were determined by the yellow phosphor-vanado-molybdate complex method [58], as previously described [47,55]. The P was extracted with 2 mL of 36% *v/v* HCl. A reagent of 5 mL of 5 M HCl and 5 ml of ammonium molybdate-ammonium metavanadate was used. Phosphorus concentration was determined by a Beckman Coulter DU 800 spectrophotometer by reading the absorbance at 400 nm, as previously described by others [47,55].

### 3.5. Cottonseed Protein and Oil Analysis

Protein and oil contents were measured in mature cottonseed. Briefly, approximately 25 g of seed was ground using a Laboratory Mill 3600 (Perten, Springfield, IL, USA). The contents of protein and oil in cottonseed were analyzed by near infrared reflectance [3,4,59] using a diode array feed analyzer AD 7200 (Perten, Springfield, IL, USA). Calibrations were developed using Perten’s Thermo Galactic Grams PLS IQ software, and the calibration equation was established according to AOAC methods AOAC, 1990a [60]; AOAC, 1990b [61]. Protein and oil were expressed on a seed dry matter basis [62,63]

## 4. Conclusions

The current research showed that curly leaf trait can alter cottonseed nutrition, including protein, oil, and some macronutrients, including N, P, K, and Ca. Higher levels of cottonseed oil in Uzbek CRL and DP 5690 wild type may provide a potential source of oil and nutrients, although the range of seed protein, oil, and all seed nutrients still fall within the normal ranges. The lower levels of protein in Uzbek CRL and DP 5690 wild type compared with DP 5690 CRL are due to the inverse relationships between protein and oil. Since the Uzbek CRL isoline was higher in oil and minerals than the DP 5690 CRL isoline (both are near isolines for curly leaf trait), curly leaf trait may be partially involved in the regulation and accumulation of oil and nutrients in cotton seeds. These constituents determine cottonseed nutritional qualities for human health and livestock feed. This research is beneficial to breeders for cotton selection for high seed oil or high protein, and to other researchers for further understanding the effect of morphological traits such as leaf shape on cottonseed protein, oil, and mineral nutrition.

## Figures and Tables

**Figure 1 plants-10-00525-f001:**
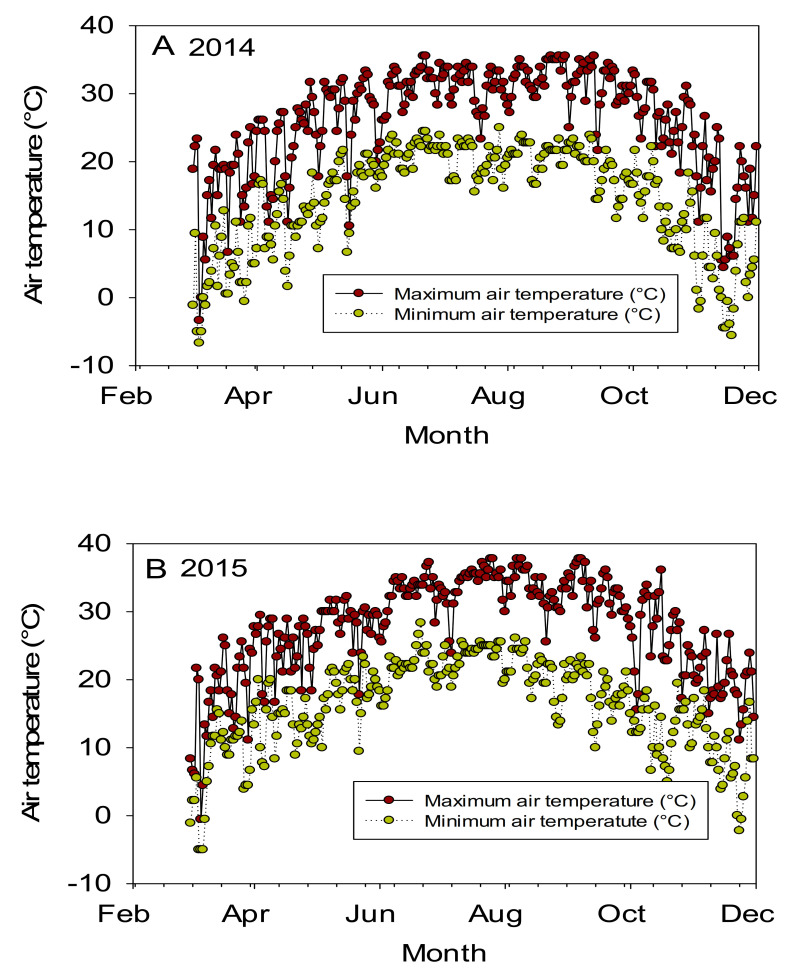
Air temperature (°C) during the growing season in 2014 (**A**) and 2015 (**B**). The experiment was conducted over 2014 and 2015 in Stoneville MS. Mississippi State University Extension, Delta Weather Center [40], verified on 30 January 2021.

**Figure 2 plants-10-00525-f002:**
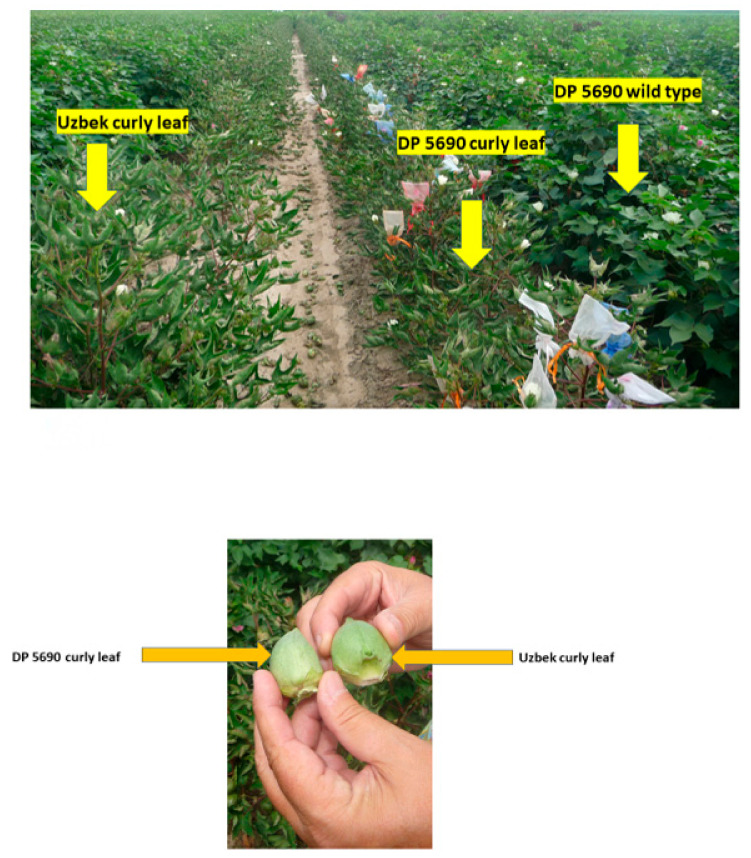
Uzbek curly leaf, DP 5690 wild type (smaller, narrow, rolled-up leaves), and DP 5690 curly leaf lines in the field, top photo; bolls from Uzbek curly leaf (larger) and DP 5690 curly leaf lines (smaller) showing the boll sizes, bottom photo.

**Table 1 plants-10-00525-t001:** Analysis of variance (F and P values) for the effect of leaf shape trait (curly leaf) on cottonseed protein and oil (g/kg); calcium (Ca), magnesium (Mg), potassium (K), phosphorus (P), sulfur (S) (mg/g); carbon (C), and nitrogen (N) (%). The experiment was conducted in 2014 and 2015 in Stoneville, MS, USA.

		**Protein**		**Oil**		**Calcium**		**Magnesium**		**Potassium**	
**Effect**	**DF**	**F Value**	**P Level**	**F Value**	**P Level**	**F Value**	**P Level**	**F Value**	**P Level**	**F Value**	**P Level**
**Year**	1	3.76	*	71.53	***	7.05	*	80.82	***	30.41	***
**Line**	2	3.77	*	74.88	***	41.99	***	18.69	***	120	***
**Year*Line**	2	4.99	*	5.36	*	0.41	ns	17.32	***	4	*
**Residuals**		61.27		76.68		0.006		0.071		0.81	
		**Phosphorus**		**Sulfur**		**Carbon**		**Nitrogen**			
**Effect**	**DF**	**F Value**	**P Level**	**F Value**	**P Level**	**F Value**	**P Level**	**F Value**	**P Level**		
**Year**	1	5.02	*	17.95	**	13.08	***	46.27	***		
**Line**	2	34.58	***	69.97	***	121.82	***	39.18	***		
**Year*Line**	2	0.6	ns	0.15	ns	2.75	ns	4.55	***		
**Residuals**		0.71		0.033		0.281		0.039			

* Significance at *p*≤ 0.05; ** significance at *p* ≤ 0.01; *** significance at *p* ≤ 0.001; ns = not significant.

**Table 2 plants-10-00525-t002:** Effects of leaf shape trait (curly leaf; CRL) on cottonseed protein and oil (g/kg); calcium (Ca), magnesium (Mg), potassium (K), phosphorus (P), sulfur (S) (mg/g); carbon (C), and nitrogen (N) (%). The experiment was conducted in 2014.

Line	Protein	Oil	Ca	K	Mg	P	S	N	C
Uzbek CRL	273	299	1.51	21.17	7.07	11.23	4.27	5.83	59.07
DP 5690 wild type	267	291	1.33	18.93	7.13	10.67	4.14	4.95	58.97
DP 5690 CRL	268	251	1.76	12.30	5.50	7.03	3.17	4.67	55.27
LSD	4.5	4.49	0.59	0.71	0.34	0.79	0.10	0.085	0.33

LSD = Least Significant Difference test, significant at the 5% level. Within each column, the difference between two values is statistically significant if it equals or exceeds the corresponding LSD.

**Table 3 plants-10-00525-t003:** Effects of leaf shape trait (curly leaf; CRL) on cottonseed protein and oil (g/kg); calcium (Ca), magnesium (Mg), potassium (K), phosphorus (P), sulfur (S) (mg/g); carbon (C), and nitrogen (N) (%). The experiment was conducted in 2015.

Line	Protein	Oil	Ca	K	Mg	P	S	N	C
Uzbek CRL	271	345	1.3	17.17	3.96	11.23	4.27	4.87	58.50
DP 5690 wild type	268	327	1.2	17.10	3.97	10.67	4.60	4.68	58.04
DP 5690 CRL	292	273	1.6	11.10	3.20	7.03	3.63	4.02	53.55
LSD	4.66	5.7	0.059	0.22	0.097	0.14	0.13	0.14	0.29

LSD = Least significant difference test, significant at the 5% level. Within each column, the difference between two values is statistically significant if it equals or exceeds the corresponding LSD.

**Table 4 plants-10-00525-t004:** Pearson correlation coefficients (P and R values) between seed nutrients in the near-isogenic lines (Uzbek curly leaf, DP 5690 wild type, and DP 5690 curly leaf) cotton in 2014. The experiment was conducted in Stoneville, MS, USA.

	Protein	Oil	Ca	K	Mg	P	S	C
**Oil**	ns							
**Ca**	ns	−0.629 *						
**K**	ns	0.903 ***	−0.663 *					
**Mg**	ns	0.827 **	ns	0.773 **				
**P**	ns	0.778 *	−0.738 *	0.880 **	ns			
**S**	ns	0.907 ***	−0.785 *	0.876 **	0.857 **	0.783 **		
**C**	ns	0.948 ***	−0.686 *	0.951 ***	0.853 **	0.779 **	0.921 ***	
**N**	ns	0.733 *	ns	0.809 **	ns	0.671 *	0.669 *	0.742 *

* Significance at *p* ≤ 0.05; ** significance at *p* ≤ 0.01; *** significance at *p* ≤ 0.001; ns = not significant.

**Table 5 plants-10-00525-t005:** Pearson correlation coefficients (P and R values) between seed nutrients in the near-isogenic lines (Uzbek curly leaf, DP 5690 wild type, and DP 5690 curly leaf) cotton in 2015. The experiment was conducted in Stoneville, MS, USA.

	Protein	Oil	Ca	K	Mg	P	S	C
**Oil**	−0.849 **							
**Ca**	ns	−0.872 **						
**K**	−0.865 **	0.930 ***	−0.861 **					
**Mg**	ns	ns	ns	ns				
**P**	−0.865 **	0.906 ***	−0.818 **	0.982 ***	ns			
**S**	−0.802 **	0.843 **	−0.826 **	0.908 ***	ns	0.940 ***		
**C**	−0.796 **	0.925 ***	−0.844 **	0.974 ***	ns	0.965 ***	0.917 ***	0.859 **
**N**	−0.853 **	0.835 **	−0.710 *	0.878 **	ns	0.887 **	0.899 ***	0.859 **

* Significance at *p* ≤ 0.05; ** significance at *p* ≤ 0.01; *** significance at *p* ≤ 0.001; ns = not significant.

## Data Availability

Not applicable.

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
