# Peer review of "Cottonseed Protein, Oil, and Minerals in Cotton (Gossypium hirsutum L.) Lines Differing in Curly Leaf Morphology"

_plants, 2021, doi:10.3390/plants10030525_

Round 1

Reviewer 1 Report

General remarks

Dear Colleagues / Researchers!

The manuscript entitled "Cottonseed protein, oil, and minerals in cotton (Gossypium hir-2 sutum L.) lines differing in curly leaf trait" by Nacer Bellaloui, Rickie B. Turley, and Salliana R. Stetina presents interesting results on the effect of cotton leaf morphology on the chemical composition of the seeds of these plants.

The authors conducted field experiments using three cotton lines: two curly leaf lines (Uzbek CRL and DP 5690 CRL) and one normal leaf (DP 5690 wild type). In addition to the genetic factor determining the shape of the leaves, the environmental factor, which was the temperature during the growing season, was also taken into account. The obtained results were properly analyzed statistically and presented in the manuscript. The presented research has great cognitive and application value. I only have doubts about the interpretation of some of the results. Well, the authors found that the content of oil and minerals is higher in the seeds of the wild line - DP 5690 (with normal leaves) and the seeds of plants with curly leaves (Uzbek line) compared to the content of these components in the seeds of the DP 5690 198 CRL line also with curly leaves. These results suggest that the feature - leaf curl is not directly related to the content of these components in the seeds. If it were so, then in the seeds of both curly-leaved lines we should observe a similar level of these ingredients, different from the level characteristic for seeds obtained from plants with normal leaves!

It also seems to me that it would be easier to observe the effects of changing the shape of the leaves under controlled conditions, e.g. in a greenhouse, because then it would be possible to completely eliminate (or control) the influence of environmental factors on the chemical composition of seeds (but this is only a suggestion in case the authors planned conduct further research in this area).

I believe that the manuscript in its current form is a valuable work and, with minor modifications, can be printed in the journal „Plats”.

Specific remarks:

The summary is too long, please specify some of the statements especially in line 20-23

I suggest changing the title to e.g. „Cottonseed protein, oil, and minerals in cotton (Gossypium hirsutum L.) lines differing in morphology of the leaves”  or „Chemical composition of cottonseeds from lines of cotton (Gossypium hirsutum L.) with normal and curly leaves” or „The content of protein, oil and minerals in cottonseed…..”

Line 33- I don't think cotton seeds are the main source of nutrients for humans and livestock. Perhaps this is true for some countries or regions where a lot of cotton is grown.

Line 65- specify what fiber do you mean? Or write a whole sentence in a different way

The manuscript requires linguistic correction.

Line 110 rewrite the sentence „showing that curly leaf  silences specific gene expression modules and genes involved in these module  contribute to regulate specific physiological function during embryo development”

Line 143-147 rewrite the sentence

Line 408- write the correct form:  HNO3

Line 417-419 Rewrite the sentence -Phosphorus concentration was determined by a Beckman Coulter DU 800 spectrophotometer by reading the absorbance at 400  nm using as previously described by others [47,55].

Line 436  rewrite the sentence: Because the Uzbek CRL isoline was higher in oil and minerals than the DP 5690 CRL  isoline (both are near isolines for curly leaf trait)…

Line: 439 rewrite  the sentence: These constituents determine cottonseed nutritional qualities and important for human health and livestock feed

Reviewer 2 Report

These mini typos should be corrected:
line 263: space before the dot, "than the other [45] ."
line 306: space too large before “carbon”          ”(mg / g);   carbon (C), ”
line 434: 2 times "in" here: "protein in in Uzbek"

In addition to the small drafting mistakes I mentioned earlier, I would also like to point out that the "Materials and Methods" should also explain the origin of the cotton variety "Uzbek curly leaf".

Round 2

Reviewer 1 Report

Dear Authors,

I am satisfied with your response to my comments / suggestions regarding your manuscript. In this form, I believe the manuscript is suitable for publication in the journal Plants. I have no other comments regarding this manuscript. I wish the authors further success in the scientific field.